# The Genus *Neoseiulus* Hughes (Acari: Phytoseiidae) in Shanxi, China [note 1]

**DOI:** 10.3390/ani13091478

**Published:** 2023-04-26

**Authors:** Yu Liu, Fang-Xu Ren, Qing-Hai Fan, Min Ma

**Affiliations:** 1College of Plant Protection, Shanxi Agriculture University, Taigu 030801, China; 2Plant Health & Environment Laboratory, Ministry for Primary Industries, Auckland 1072, New Zealand; 3State Key Laboratory for Biology of Plant Diseases and Insect Pests, Beijing 100193, China

**Keywords:** Mesositgmata, taxonomy, predatory mite, new record

## Abstract

**Simple Summary:**

Phytoseiid mites are widely distributed on plants and in soil; they can prey on many phytophagous mites and pests and play an important role in biological control programs. The genus *Neoseiulus* Hughes is one of the largest genera within the Phytoseiidae family, comprising 14.6% of the family’s species worldwide. At present, there are few reports investigating *Neoseiulus* from Shanxi Province; however, the study on the species diversity of *Neoseiulus* is helpful to enrich the resource species bank of Phytoseiidae and provide more detailed basis for species identification. We report the discovery of five additional species in Shanxi, and redescribed four of them. *Neoseiulus paraki* (Ehara) is recorded for the first time in China, and *N. neoreticuloides* (Liang and Hu) is considered a new junior synonym of *N. bicaudus* (Wainstein). We provide a key to assist in the identification of the known species of *Neoseiulus* in Shanxi.

**Abstract:**

The genus *Neoseiulus* in Shanxi Province is reviewed and seven species are recorded from the province. Four of these are redescribed and detailed taxonomic information are provided. *Neoseiulus paraki* (Ehara) is recorded for the first time in China and *Neoseiulus neoreticuloides* (Liang and Hu) is considered a new junior synonym of *Neoseiulus bicaudus* (Wainstein). Additionally, a diagnostic key to the known species of *Neoseiulus* in Shanxi is provided.

## 1. Introduction

The genus *Neoseiulus* Hughes [1] is one of the largest genera within the Phytoseiidae family, comprising 361 (14.6%) of the family’s species worldwide [2,3,4]. In China, it is also a big genus, with 57 recorded species, accounting for 17.1% of the Phytoseiidae species in the country [3,5,6,7].

Shanxi is a province located in the northern part of China and covers a total area of 156,700 square kilometers. The majority of the province, over two-thirds, sits on the loess plateau. The climate is characterized as semiarid, with lower annual rainfall and longer dry seasons. In the south, the monthly 24 h average temperature ranges from −3 °C to 32 °C, while in the north, it ranges from −9.8 °C to 21.9 °C. The annual precipitation falls between 400 mm and 650 mm, with over 70% of the rainfall occurring between June and September. The most common natural vegetation consists of shrubs and grasses, and forested areas, mainly found on mountain slopes, only cover approximately 20% of the land area. Due to these factors, Shanxi has a lower biodiversity in comparison to areas located in southern China.

The knowledge of *Neoseiulus* species in Shanxi is limited. Currently, only two species have been recorded, namely *N. womersleyi* (Schicha) [8] (previously identified as *N. pseudolongispinosus*) [6,9,10,11] and *N. zwoelferi* Dosse [12]. The objective of this study is to report the discovery of five additional species in Shanxi, clarify the identity of *N. neoreticuloides* [13], and provide redescriptions of *N. bicaudus* (Wainstein) [14], *N. lushanensis* (Zhu and Chen) [15], *N. paraki* (Ehara) [16,17] and *N. tauricus* (Livshitz and Kuznetsov) [18]. Additionally, this paper aims to provide a key to assist in the identification of the known species of *Neoseiulus* in Shanxi.

## 2. Materials and Methods

The mites were collected using the beating method. The whole plant or branches of plants were beaten with a stick over a black rectangular plastic plate. Specimens were picked up with a fine soft hairbrush and kept in absolute ethanol before being taken to the laboratory. The mites were cleared and macerated in Nesbitt’s fluid until they became translucent, then transferred to distilled water for 2–3 min to dissolve the Nesbitt’s fluid before being mounted in Hoyer’s medium on slides under a dissecting microscope (Zeiss DV4, Carl Zeiss Microscopy GmbH, Gottingen, Germany and Optec SZ650, Chongqing Optec Instrument Co., Ltd., Chongqing, China). Using a compound microscope (Nikon Eclipse 80i, Nikon Corporation, Tokyo, Japan) with differential interference contrast (DIC), the specimens were examined, measured, and photographed. Illustrations were created using a drawing tube (Nikon Y-IDT, Nikon Corporation, Tokyo, Japan) attached to the microscope and were edited with Photoshop CC2018. The voucher specimens are deposited in the Entomological Microscopy Laboratory of College of Plant Protection, Shanxi Agricultural University.

Measurements were taken along the midline from the anterior to posterior margins for the length of the dorsal shield, sternal shield, epigynal shield and ventrianal shield. The width of the dorsal shield was measured at the level of *s4*, the sternal shield (or sternogenital in males) at the level of *st2*, the genital shield at the level of *st5*, and the ventrianal shield at the level of *ZV2*. All measurements were presented in micrometers (μm) for the specimen used for illustration and other specimens in parentheses. The terminology of the idiosomal and leg chaetotaxy, and pore-like structures follow that of Rowell et al. (1978) [19] and Chant and McMurtry (2007) [2], Evans (1963) [20], and Athias-Henriot (1975) [21], respectively. The setal pattern system of idiosoma follows that of Chant and Yoshida-Shaul (1992) [22].

## 3. Results


***Neoseiulus* Hughes, 1948**


Type species: *Neoseiulus barkeri* Hughes, 1948: 141.

### 3.1. Redescriptions of Species

#### 3.1.1. *Neoseiulus bicaudus* (Wainstein, 1962) (Figure 1, Figure 2, Figure 3, Figure 4 and Figure 5)

*Amblyseius bicaudus* Wainstein, 1962: 146.

*Typhlodromus bicaudus* (Wainstein), Hirschmann 1962: 2 [23].

*Amblyseius* (*Amblyseius*) *bicaudus* (Wainstein), Ehara, 1966: 20 [24].

*Neoseiulus bicaudus* (Wainstein), Congdon, 2002: 23 [25]; Wang et al., 2015: 456 [26].

*Amblyseius neoreticuloides* Liang and Hu, 1988: 317 [13].

*Amblyseius* (*Neoseiulus*) *neoreticuloides* (Liang and Hu), Wu et al., 2009: 105 [6].

*Neoseiulus neoreticuloides* (Liang and Hu). **New synonym**.

**Diagnosis (female)**. Dorsal shield elongate oval, reticulated throughout, bearing 17 pairs of setae, 16 pairs of lyrifissures and 7 pairs of solenostomes, *S4*, *S5*, *Z4* and *Z5* serrated, others smooth; *Z5* longer than others. Peritremes extending anteriorly to level of *j1*. Sternal shield reticulated, bearing three pairs of setae. Ventrianal shield approximately pentagonal, reticulated; solenostomes (*gv3*) posteromedian to *JV2*, circular. Calyx of spermathecal apparatus cup-shaped and basally stalked, and stalk approximately as long as width of atrium; atrium nodular at junction with minor duct, minor duct thread-like for a short distance and then expanded, forming a cylindrical tube. Fixed digit of chelicera with six teeth, movable digit with a tooth. Palpgenu with genu setae *al1* and *al2* rod-like. Leg genu II with seven setae. Only basitarsus of leg IV with a macroseta.**Redescription. Female** (n = 7). Dorsal idiosoma (Figure 1A and Figure 2A). Idiosomal setal pattern 10A:9B/JV-3:ZV. Dorsal shield elongate oval, fully reticulate, has a waist located slightly below *R1*, 394 (380–412) long, 183 (181–189) wide; muscle marks visible between *j3* and *Z4*, a pair of muscle marks present in front of *J5*. Dorsum with 17 pairs of setae and 16 pairs of lyrifissures (*id1*, *id2*, *id4*, *id6*, *idx*, *idx1*, *idl2*, *idl3*, *idl4*, *idm1*, *idm2*, *idm3*, *idm4*, *idm5*, *idm6* and *is1*) and 7 pairs of solenostomes (*gd1*, *gd2*, *gd4*, *gd5*, *gd6*, *gd8* and *gd9*); lyrifissures *id3* and solenostomes *gd3* on peritremal shield. All dorsal setae smooth except for serrated *S4*, *S5*, *Z4* and *Z5*; *Z5* longer than others. Lateral setae *r3* and *R1* smooth, on interscutal membrane. Peritremes extending anteriorly close to *j1*, posterior part of peritremal shield (Figure 1B) curved and pointed, protuberance of exopodal shield at level of the stigmata. Lengths of setae: *j1* 22 (22–24), *j3* 28 (27–31), *j4* 13 (12–14), *j5* 13 (11–14), *j6* 16 (15–16), *J2* 18 (15–19), *J5* 12 (12–14), *r3* 27 (26–30), *R1* 25 (24–26), *s4* 31 (27–33), *S2* 29 (29–32), *S4* 34 (32–36), *S5* 41 (39–44), *z2* 22 (19–27), *z4* 20 (16–21), *z5* 11 (11–13), *Z1* 20 (19–22), *Z4* 35 (34–41), *Z5* 80 (80–95).

**Figure 1 animals-13-01478-f001:**
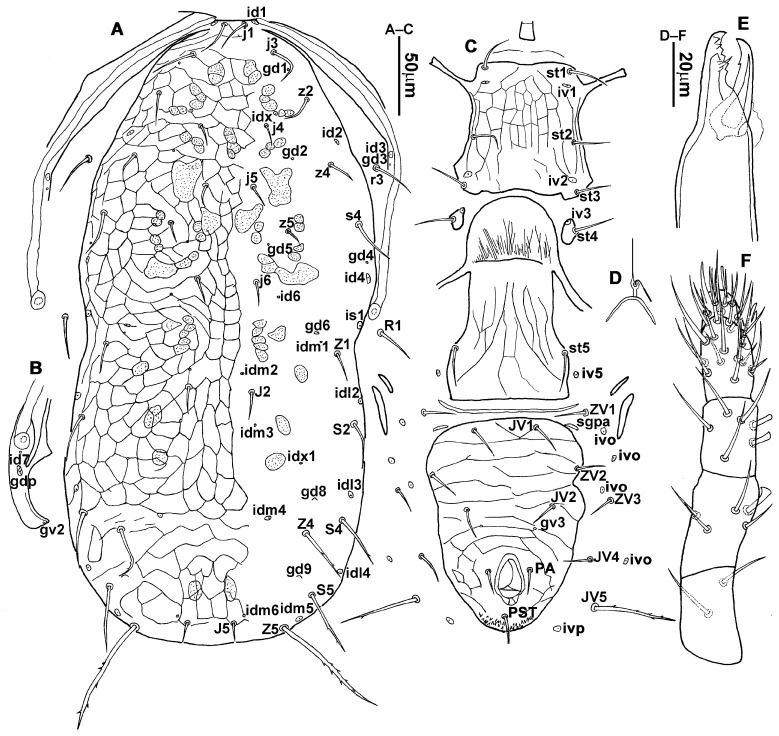
*Neoseiulus bicaudus* (Wainstein), female. (**A**) Dorsal shield; (**B**) Posterior part of peritremal shield and exopodal shield; (**C**) Ventral idiosoma; (**D**) Spermathecal apparatus; (**E**) Chelicera; (**F**) Palp.

3.Ventral idiosoma (Figure 1C and Figure 2B). Sternal shield reticulated, 79 (77–83) long, 74 (74–77) wide; anterior margin laterally convex, forming a flat M-shaped projection, posterior margin concave, arched above the level of bases of *st3*; three pairs of setae (*st1*, *st2* and *st3*) and two pairs of lyrifissures (*iv1* and *iv2*) present on sternal shield, *iv1* positioned posteriad of *st1*, *iv2* positioned between *st2* and *st3*, and close to *st3*. Metasternal platelets small, each bearing a seta *st4* and a lyrifissure *iv3*. Epigynal shield slightly striated, 132 (125–136) long, 76 (73–76) wide. Lengths of setae: *st1* 25 (24–29), *st2* 24 (21–24), *st3* 23 (21–25), *st4* 24 (22–27), *st5* 24 (20–25). Approximately four slender transverse sclerites present between epigynal and ventrianal shields. Ventrianal shield approximately pentagonal, transversally striated and a few oblique striae present between transverse striae, 136 (134–142) long, 100 (97–108) wide, bearing 3 pairs of preanal setae (*JV1*, *JV2* and *ZV2*), a pair of paranal setae (*PA*) and a postanal seta (*PST*), and a pair of circular solenostomes (*gv3*) posteromedial to *JV2*, distance *gv3*–*gv3* 36 (36–43); 4 pairs of setae (*JV4*, *JV5*, *ZV1* and *ZV3*) and 5 pairs of lyrifissures present on soft cuticle surrounding ventrianal shield, *JV5* serrate, others smooth. A pair of tiny platelets (*sgpa*) posteroparaxial to *ZV1* adjacent to anterior corners of ventrianal shield. Lengths of setae: *JV1* 18 (17–20), *JV2* 17 (16–18), *JV4* 17 (16–20), *JV5* 52 (52–60), *ZV1* 18 (18–20), *ZV2* 18 (17–19), *ZV3* 15 (15–17). Primary metapodal platelet 32 (30–35) long, 4 (4–6) wide; secondary platelet 15 (12–17) long, 2 (2–3) wide.4.Spermatheca (Figure 1D and Figure 3A,B). Calyx of spermathecal apparatus cup-shaped and basally stalked, 11 (10–12) long; stalk approximately as long as width of atrium, atrium nodular at junction with minor duct; minor duct thread-like for a short distance and then expanded, forming a cylindrical tube; major duct slender.5.Gnathosoma. Chelicera (Figure 1E and Figure 2C) with fixed digit 37 (33–37) long, bearing six teeth, pilus dentilis located at the level of fourth tooth, 6 (6–8) long; movable digit 32 (31–34) long, bearing single tooth. Palp (Figure 1F). Trochanter with two simple setae; femur with a spatulate and four simple setae; genu bearing two rod-like setae (*al1* and *al2*) and four simple setae; tarsal apotele two-tined.6.Legs (Figure 4A–D). Leg I 357 (341–362) long, setal formula: coxa 0-0/1-0/1-0, trochanter 1-0/1-0/2-1, femur 2-3/1-2/2-2, genu 2-2/1-2/1-2, tibia 2-2/1-2/1-2, basitarsus 0-0/0-1/0-0. Apical sensorial setal cluster of tarsus I (Figure 4E) with nine modified setae. Leg II 300 (290–303) long, setal formula: coxa 0-0/1-0/1-0, trochanter 1-0/1-0/2-1, femur 2-3/1-2/1-1, genu 2-2/0-2/0-1, tibia 1-1/1-2/1-1, basitarsus 1-1/0-1/0-1. Leg III 307 (295–307) long, setal formula: coxa 0-0/1-0/1-0, trochanter 1-1/1-0/2-0, femur 1-2/1-1/0-1, genu 1-2/1-2/0-1, tibia 1-1/1-2/1-1, basitarsus 1-1/0-1/0-1. Leg IV 400 (386–404) long, setal formula: coxa 0-0/1-0/0-0, trochanter 1-1/1-0/2-0, femur 1-2/1-1/0-1, genu 1-2/1-2/0-1, tibia 1-1/1-2/0-1, basitarsus 1-1/0-1/0-1. Legs I–III without obvious macrosetae; leg IV with a smooth macroseta on basitarsus, 73 (73–78) long.7.**Male** (n = 4). Dorsal idiosoma (Figure 5A). Dorsal shield elongate oval, presenting distinct reticulation throughout, 312 (301–312) long, 163 (153–166) wide; muscle marks visible between *j3* and *J5*; dorsum bearing 19 pairs of setae, 16 pairs of lyrifissures (*id1*, *id2*, *id4*, *id6*, *idx*, *idx1*, *idl2*, *idl3*, *idl4*, *idm1*, *idm2*, *idm3*, *idm4*, *idm5*, *idm6* and *is1*) and 7 pairs of solenostomes (*gd1*, *gd2*, *gd4*, *gd5*, *gd6*, *gd8* and *gd9*). Setae *S5*, *Z4* and *Z5* serrate, others smooth; *Z4* and *Z5* longer than others. Peritremes anteriorly ending at level between *j1* and *j3*. Lengths of setae: *j1* 21 (19–21), *j3* 24 (24–27), *j4* 14 (13–14), *j5* 13 (13–15), *j6* 16 (14–17), *J2* 19 (17–19), *J5* 12 (10–12), *r3* 27 (24–27), *R1* 23 (22–26), *s4* 29 (29–32), *S2* 31 (30–32), *S4* 34 (32–34), *S5* 36 (34–39), *z2* 20 (17–20), *z4* 15 (15–20), *z5* 14 (12–14), *Z1* 19 (19–21), *Z4* 43 (43–46), *Z5* 69 (67–69).8.Ventral idiosoma (Figure 5B). Sternogenital shield sparsely striated between *st1* and *st4*, reticulate between *st3* and *st4*, 133 (132–137) long, 61 (60–63) wide; anterior margin prominently convex, posterior margin nearly straight, bearing five pairs of setae (*st1*, *st2*, *st3*, *st4* and *st5*) and three pairs of lyrifissures (*iv1*, *iv2* and *iv3*); lengths of setae: *st1* 23 (19–23), *st2* 18 (17–18), *st3* 17 (16–18), *st4* 18 (17–18), *st5* 19. Ventrianal shield subtriangular, reticulate throughout, 122 (114–122) long, 133 (127–133) wide; with three pairs of preanal setae (*JV1*, *JV2* and *ZV2*), a pair of paranal setae (*PA*) and a postanal seta (*PST*), three pairs of lyrifissures, a pair of circular solenostomes (*gv3*) posteromesad to *JV2*, distance *gv3*–*gv3* 28 (27–32), two pairs of marginal muscle marks situated anterolateral to anus. Setae *JV5* serrate, and a pair of lyrifissures on soft cuticle surrounding ventrianal shield. Lengths of setae: *JV1* 15 (14–15), *JV2* 17 (16–17), *JV5* 44 (41–45), *ZV2* 19 (18–20).

**Figure 2 animals-13-01478-f002:**
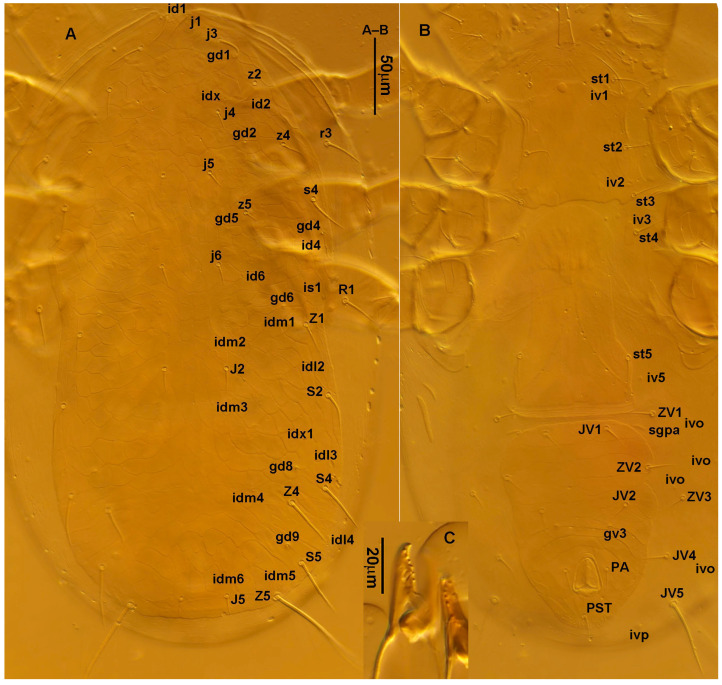
*Neoseiulus bicaudus* (Wainstein), female. (**A**) Dorsal shield; (**B**) Ventral idiosoma; (**C**) Chelicera.

9.Gnathosoma. Chelicera (Figure 5C) with fixed digit 24 (22–24) long, bearing four teeth, movable digit 23 (21–23) long, bearing a tooth. Spermatodactyl L-shaped, with acute toe and heel, shaft 16 (16–17) long, foot 7 (6–7) long. Palp and hypostome with same chaetotaxy as in female.10.Legs. Leg chaetotaxy same as those in adult female.

**Figure 3 animals-13-01478-f003:**
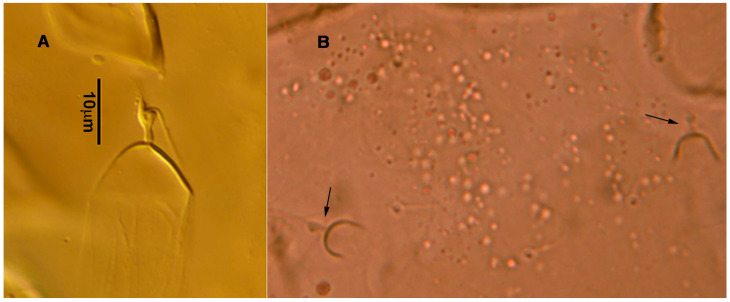
*Neoseiulus bicaudus* (Wainstein), female spermathecal apparatus: (**A**) Shanxi specimen; (**B**) Holotype of *N. neoreticuloides* (The spermathecal apparatuses are marked with arrows) (from Weinan Wu).

**Figure 4 animals-13-01478-f004:**
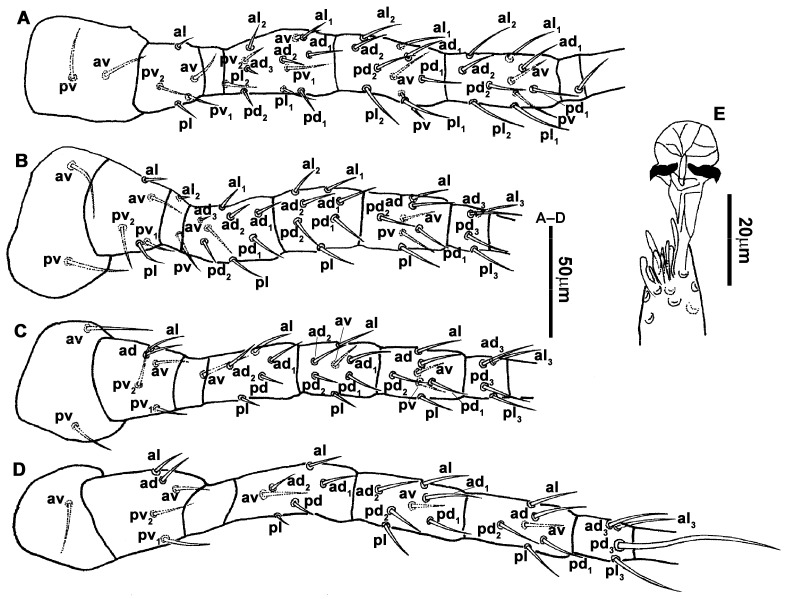
*Neoseiulus bicaudus* (Wainstein), female. (**A**) Leg I; (**B**) Leg II; (**C**) Leg III; (**D**) Leg IV; (**E**) Apical sensorial setal cluster of tarsus I.

11.**Materials examined.** A total of 10♀ and 1♂, Dabaishi Village, Taigu County, Shanxi Province, 37°20′12″ N, 112°38′50″ E, 1340 m, e.g., *Setaria viridis* (L.) P. Beauv. (Poaceae), 31 August 2020, Y. Liu, M. Ma, B. Zhang and F.-X. Ren coll.; 13♀ and 1♂, Shanxi Agriculture University, Taigu County, Shanxi Province, 37°25′15″ N, 112°34′37″ E, 794 m, *Hemerocallis fulva* (L.) L. (Asphodelaceae), 7 October 2013, Qing-Hai Fan coll. (accession no.: T13_0009); 8♀ and 1♂, same locality and host as T13_0009, 2 July 2016, M. Ma coll.; 1♀, same locality and host as T13_0009, 5 July 2016, M. Ma coll. (T16_0002); 1♀ and 1♂, locality and host as T13_0009, 27 June 2016, Y.-X. Li coll.; 9♀, same locality and host as T13_0009, but 28 June 2016, M. Ma coll.; 4♀ and 1♂, same collection locality and host as T13_0009, 10 October 2013, M. Ma and Y.-N. Zhao coll. (T13_0017); 8♀ and 3♂, same locality and host as T13_0009, 20 September 2014, M.-J. Yi and B.-Q. Su coll. (T14_0299).

**Figure 5 animals-13-01478-f005:**
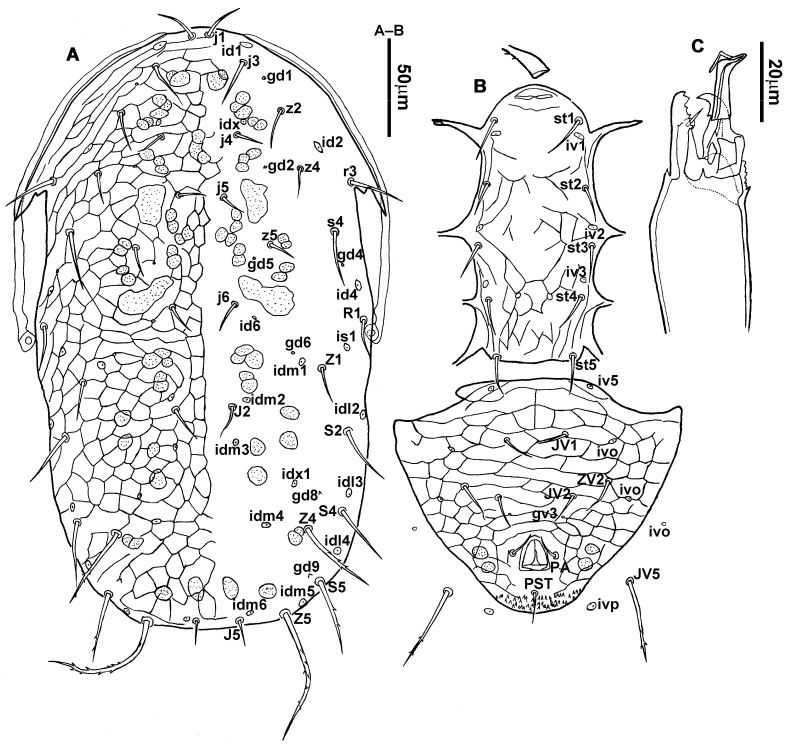
*Neoseiulus bicaudus* (Wainstein), male. (**A**) Dorsal shield; (**B**) Ventral idiosoma; (**C**) Chelicera.

12.**Remarks.** This species was originally described from grass found in Kazakhstan [14]. Subsequently, it has been collected from various hosts, including the following: almond trees [27], *Cichorium intybus* [28]; *Citrus* spp., *Cydonia oblonga*, *Fragaria ananassa*, *Malus communis*, *Prunus avium*, *Prunus domestica*, *Prunus persica* [29]; *Cupressus* sp. [30]; *Cynodon dactylon*, *Phoenix dactylifera* [31]; *Populus nigra* var. *thevestina* (Dode) bean [26]; pussy willow [32]; raspberry, low-growing vegetation [25]; *Setaria macrostachya*, *Phalaris minor*, *Cynodon dactylon*, *Pinus strobus*, *Aster spinosus*, *Haplopappus*, *Peyanum mexicanum*, *Rhynchelytrum repens*, *Asclepias curassavica*, *Distichilis stricta* [33]; *Setaria viridis* (present paper); *Tropaeolum majus* [34]; *Ulmus pumila* [13]; *Vitis vinifera* [35] in the Nearctic, Neotropical and Palearctic realms [3]. In China, it has been recorded as *Neoseiulus neoreticuloides* from Ningxia (Yinchuan) [13], and Xinjiang Uygur Autonomous Region [26].13.The type specimen of *Neoseiulus neoreticuloides* was an adult female collected from *Ulmus pumila* (Siberian elm) in Ningxia. It should be noted that the host information was mistakenly attributed to *Platycladus orientalis* in the English abstract of the same publication, which actually pertains to a different species, *Typhlodromus* (*Anthoseius*) *yinchuanensis*, by Liang and Hu. The specimens collected from Shanxi are consistent with the description provided by Liang and Hu (1988) [13] and Wu et al. (2009) [6]. Based on the presence of a stalk between the calyx and atrium of the spermatheca, *N. neoreticuloides* should be assigned to the *paraki* species subgroup. After comparing *N. neoreticuloides* with *N. bicaudus* (Table 1), we did not find any significant differences. Therefore, based on our observations, we consider *N. neoreticuloides* to be conspecific with *N. bicaudus*. As *N. bicaudus* has taxonomic priority, we conclude that *N. neoreticuloides* is a junior synonym of *N. bicaudus*.

#### 3.1.2. *Neoseiulus lushanensis* (Zhu and Chen, 1985) (Figure 6, Figure 7, Figure 8, Figure 9, Figure 10 and Figure 11)

*Amblyseius lushanensis* Zhu and Chen, 1985: 273.

*Amblyseius* (*Neoseiulus*) *lushanensis* (Zhu and Chen), Wu et al. 2009: 152 [6].

*Amblyseius* (*Amblyseius*) *longisiphonulus* Wu and Lan, 1989: 248 [36]; synonym by Wu et al. 2009: 152.

*Neoseiulus lushanensis* (Zhu and Chen), Chant and McMurtry, 2003: 37 [37]; Moraes et al. 2004: 131 [38].

**Diagnosis (female)**. Dorsal shield oval, mostly smooth but striated anterolaterally, bearing 17 pairs of setae, 16 pairs of lyrifissures and 7 pairs of solenostomes, all smooth but *Z4* and *Z5* serrate; *j3*, *S2*, *s4*, *z2*, *z4*, *Z4* and *Z5* longer than others. Peritremes extending anteriorly close to base of *j1*. Sternal shield obviously wider than long, sparsely striated, bearing three pairs of setae. Ventrianal shield approximately pentagonal, transversally striated and a few oblique striae present between transverse striae, with three pairs of preanal setae; solenostomes (*gv3*) posteromedian to *JV2*, crescent-shaped. Calyx of spermathecal apparatus elongate trumpet-shaped, arms distally thickened and flaring distally, atrium very large, bifurcate at junction with major duct, major duct membranous and thick-walled, as broad as atrium and then gradually reduced. Fixed digit of chelicera with five–six teeth and movable digit with one tooth. Palpgenual setae *al1* and *al2* rod-like. Leg genu II with 7 setae. Genu, tibia and basitarsus of leg IV each with a macroseta.

**Figure 6 animals-13-01478-f006:**
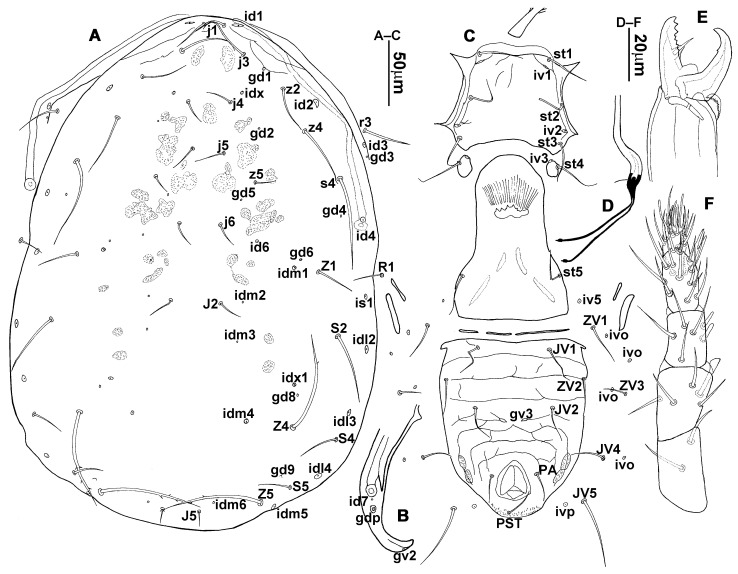
*Neoseiulus lushanensis* (Zhu and Chen), female. (**A**) Dorsal shield; (**B**) Posterior part of peritremal shield and exopodal shield; (**C**) Ventral idiosoma; (**D**) Spermathecal apparatus; (**E**) Chelicera; (**F**) Palp.

2.**Redescription. Female** (n = 7). Dorsal idiosoma (Figure 6A and Figure 7A). Idiosomal setal pattern 10A:9B/JV-3:ZV. Dorsal shield oval, with a waist at level of *R1*; 399 (373–411) long, 262 (242–271) wide; shield mostly smooth but with a few striae at anterolateral margins between *j1* and *z2*; muscle marks visible between *j3* and *Z4*. Dorsum with 17 pairs of setae, 16 pairs of lyrifissures (*id1*, *id2*, *id4*, *id6*, *idx*, *idx1*, *idl2*, *idl3*, *idl4*, *idm1*, *idm2*, *idm3*, *idm4*, *idm5*, *idm6* and *is1*) and 7 pairs of solenostomes (*gd1*, *gd2*, *gd4*, *gd5*, *gd6*, *gd8* and *gd9*); lyrifissures *id3* and solenostomes *gd3* on peritremal shield. Dorsal setae *Z4* and *Z5* serrated, others smooth, *j3*, *s4*, *S2*, *z2*, *z4*, *Z4* and *Z5* longer than others. Lateral setae *r3* and *R1* smooth, on interscutal membrane. Peritremes extending anteriorly close to bases of *j1*, posterior part of peritremal shield (Figure 6B) curved and pointed, protuberance of exopodal shield in front of stigmata. Lengths of setae: *j1* 30 (22–30), *j3* 50 (44–52), *j4* 19 (18–20), *j5* 20 (16–23), *j6* 23 (18–23), *J2* 21 (17–24), *J5* 10 (9–10), *r3* 36 (32–37), *R1* 21 (20–24), *s4* 77 (71–80), *S2* 58 (52–65), *S4* 34 (26–37), *S5* 23 (18–27), *z2* 39 (34–42), *z4* 48 (43–49), *z5* 14 (12–14), *Z1* 35 (28–36), *Z4* 82 (74–82), *Z5* 83 (70–83).

**Figure 7 animals-13-01478-f007:**
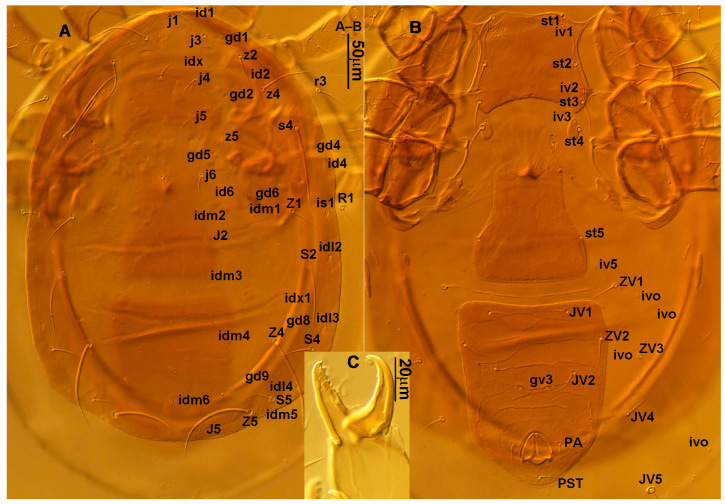
*Neoseiulus lushanensis* (Zhu and Chen), female. (**A**) Dorsal idiosoma; (**B**) Ventral idiosoma; (**C**) Chelicera.

3.Ventral idiosoma (Figure 6C and Figure 7B). Sternal shield mostly smooth, striated laterally, 66 (65–69) long, 82 (82–88) wide; anterior margin weakly convex, forming a weak M-shaped median projection; posterior margin obviously concave, arched above the level of bases of *st3*; three pairs of setae (*st1*, *st2* and *st3*) and two pairs of lyrifissures (*iv1* and *iv2*) present, *iv1* positioned posteriad of *st1*, *iv2* positioned between *st2* and *st3*, and close to *st3*. Metasternal platelets small, each bearing a seta *st4* and a lyrifissure *iv3*. Epigynal shield smooth, 134 (127–139) long, 85 (78–85) wide, with two pairs of muscle scars between *st5*–*st5*. Lengths of setae: *st1* 29 (29–37), *st2* 31 (31–37), *st3* 31 (30–36), *st4* 29 (29–37), *st5* 25 (25–32). A slender transverse sclerite (sometimes broken into three or four parts) present between epigynal and ventrianal shields. Ventrianal shield approximately pentagonal, 134 (124–144) long, 119 (114–126) wide, transversally striated and a few oblique striae present between transverse striae, bearing three pairs of preanal setae (*JV1*, *JV2* and *ZV2*), a pair of paranal setae (*PA*) and a postanal seta (*PST*), and a pair of obviously crescent-shaped solenostomes (*gv3*) posteromedial to *JV2*, distance *gv3*–*gv3* 21 (17–30); two pairs of marginal muscle marks situated anterolateral to anus; four pairs of setae (*JV4*, *JV5*, *ZV1* and *ZV3*) and five pairs of lyrifissures present on soft cuticle surrounding ventrianal shield. Lengths of setae: *JV1* 24 (23–30), *JV2* 28 (21–28), *JV4* 26 (20–26), *JV5* 59 (46–62), *ZV1* 29 (23–31), *ZV2* 25 (25–29), *ZV3* 21 (17–21). Primary metapodal platelet 29 (26–31) long, 5 wide; secondary platelet 12 (12–15) long, 2 wide.

**Figure 8 animals-13-01478-f008:**
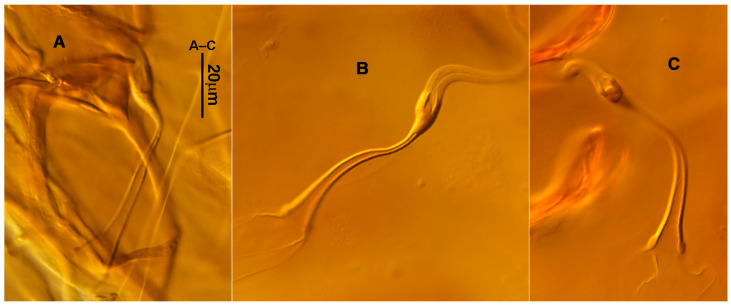
*Neoseiulus lushanensis* (Zhu and Chen), female. (**A**–**C**) Variation in spermathecal apparatus.

**Figure 9 animals-13-01478-f009:**
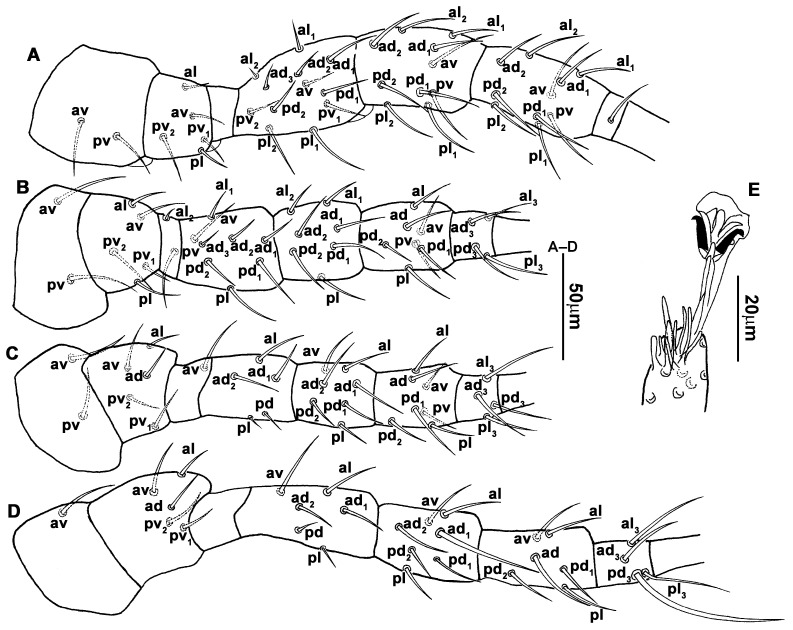
*Neoseiulus lushanensis* (Zhu and Chen), female. (**A**) Leg I; (**B**) Leg II; (**C**) Leg III; (**D**) Leg IV; (**E**) Apical sensorial setal cluster of tarsus I.

4.Spermatheca (Figure 6D and Figure 8A–C). Calyx of spermathecal apparatus elongate trumpet-shaped, distally thickened and flaring, 50 (47–51) long; a large atrium nodular at base of calyx and without a neck, bifurcate at junction with major duct; major duct membranous and thick-walled, as wide as atrium and then gradually reduced.5.Gnathosoma. Chelicera (Figure 6E and Figure 7C) with fixed digit 41 (38–41) long, bearing five–six teeth, pilus dentilis located at the level of tooth six, 7 (7–9) long, movable digit 37 (33–38) long, bearing a single tooth. Palp (Figure 6F). Trochanter with two setae; femur with a spatulate and four simple setae; genu bearing two rod-like setae (*al1* and *al2*) and four simple setae; tarsal apotele two-tined.6.Legs (Figure 9A–D). Leg I 416 (413–433) long, setal formula: coxa 0-0/1-0/1-0, trochanter 1-0/1-0/2-1, femur 2-3/1-2/2-2, genu 2-2/1-2/1-2, tibia 2-2/1-2/1-2, basitarsus 0-0/0-1/0-0. Apical sensorial setal cluster (Figure 9E) with nine modified setae. Leg II 333 (323–334) long, setal formula: coxa 0-0/1-0/1-0, trochanter 1-0/1-0/2-1, femur 2-3/1-2/1-1, genu 2-2/0-2/0-1, tibia 1-1/1-2/1-1, basitarsus 1-1/0-1/0-1. Leg III 325 (318–329) long, setal formula: coxa 0-0/1-0/1-0, trochanter 1-1/1-0/2-0, femur 1-2/1-1/0-1, genu 1-2/1-2/0-1, tibia 1-1/1-2/1-1, basitarsus 1-1/0-1/0-1. Leg IV 445 (436–457) long, setal formula: coxa 0-0/1-0/0-0, trochanter 1-1/1-0/2-0, femur 1-2/1-1/0-1, genu 1-2/1-2/0-1, tibia 1-1/1-2/0-1, basitarsus 1-1/0-1/0-1. Legs I–III without macrosetae. Genu, tibia and basitarsus of leg IV each with a smooth macroseta, *SgeIV* 56 (50–59), *StiIV* 36 (33–38) and *StIV* 81 (76–86).

**Figure 10 animals-13-01478-f010:**
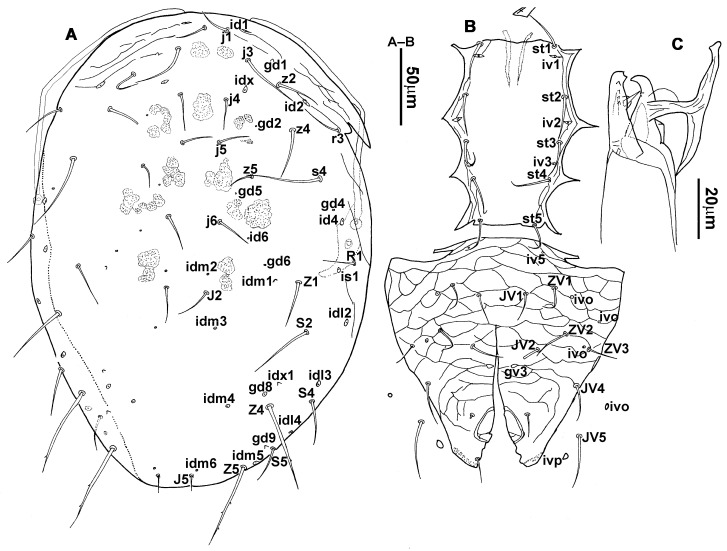
*Neoseiulus lushanensis* (Zhu and Chen), male. (**A**) Dorsal shield; (**B**) Ventral idiosoma; (**C**) Chelicera.

7.**Male** (n = 4). Dorsal idiosoma (Figure 10A and Figure 11A). Dorsal shield oval, 308 (306–309) long, 216 (211–216) wide; mostly smooth but striated anterolaterally; muscle marks visible between *j3* and *J2*; dorsum bearing 19 pairs of setae, 16 pairs of lyrifissures (*id1*, *id2*, *id4*, *id6*, *idx*, *idx1*, *idl2*, *idl3*, *idl4*, *idm1*, *idm2*, *idm3*, *idm4*, *idm5*, *idm6* and *is1*) and 7 pairs of solenostomes (*gd1*, *gd2*, *gd4*, *gd5*, *gd6*, *gd8* and *gd9*). *Z4* and *Z5* slightly serrated, other setae smooth; *j3*, *s4*, *S2*, *z2*, *z4*, *Z4* and *Z5* longer than others. Peritremes extending anteriorly close to *j1*. Lengths of setae: *j1* 21 (21–23), *j3* 41 (38–42), *j4* 19 (19–21), *j5* 19 (19–23), *j6* 23 (21–23), *J2* 21 (19–21), *J5* 9 (7–11), *r3* 30 (26–30), *R1* 22 (20–24), *s4* 56 (51–56), *S2* 48 (43–48), *S4* 32 (30–32), *S5* 28 (22–29), *z2* 33 (29–33), *z4* 28 (28–37), *z5* 14 (13–15), *Z1* 26 (26–32), *Z4* 58 (58–62), *Z5* 56 (56–63).

**Figure 11 animals-13-01478-f011:**
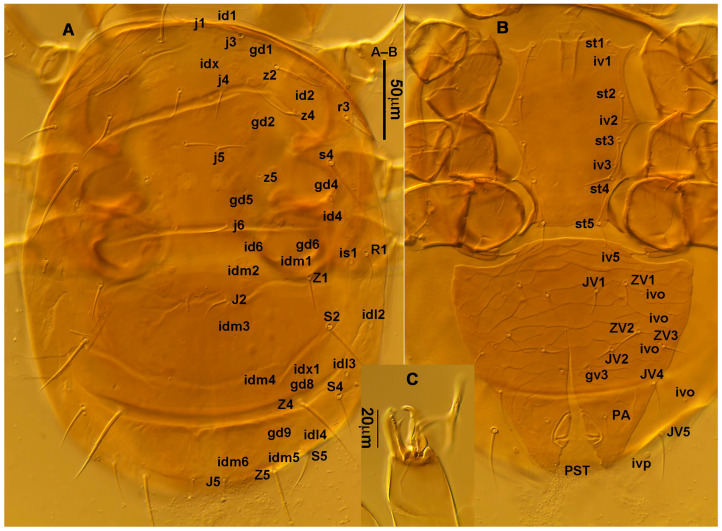
*Neoseiulus lushanensis* (Zhu and Chen), male. (**A**) Dorsal idiosoma; (**B**) Ventral idiosoma; (**C**) Chelicera.

8.Ventral idiosoma (Figure 10B and Figure 11B). Sternogenital shield mostly smooth but a few striae present at lateral margins, 122 (120–124) long, 77 (72–77) wide; anterior margin and posterior margin nearly straight, bearing five pairs of setae (*st1*, *st2*, *st3*, *st4* and *st5*) and three pairs of lyrifissures (*iv1*, *iv2* and *iv3*); lengths of setae: *st1* 26 (26–31), *st2* 25 (25–30), *st3* 27 (27–30), *st4* 22 (22–29), *st5* 25 (23–25). Ventrianal shield subtriangular, reticulated throughout, anterior margin convex medially, 152 (134–152) long, 144 (141–144) wide, with six pairs of preanal setae (*JV1*, *JV2*, *JV4*, *ZV1*, *ZV2* and *ZV3*), a pair of paranal setae (*PA*) and a postanal seta (*PST*); four pairs of lyrifissures on ventrianal shield, a pair of crescent-shaped solenostomes (*gv3*) posteromedian to *JV2*, distance *gv3*–*gv3* 16 (16–22). Setae *JV5* and three pairs of lyrifissures on soft cuticle surrounding ventrianal shield. Lengths of setae: *JV1* 19 (19–25), *JV2* 20 (20–25), *JV4* 23 (21–23), *JV5* 33 (31–39), *ZV1* 18 (18–25), *ZV2* 22 (22–26), *ZV3* 17 (17–19).9.Gnathosoma. Chelicera (Figure 10C and Figure 11C) with fixed digit 28 (28–31) long, bearing five teeth; movable digit 26 (26–29) long, bearing a tooth. Spermatodactyl T-shaped, with acute toe and heel, shaft 22 (20–22) long, foot 47 (46–49) long. Palp and hypostome with same chaetotaxy as in female.10.Legs. Leg chaetotaxy same as that in adult female.11.**Materials examined.** A total of 2♀ and 2♂, Dabaishi Village, Taigu County, Shanxi Province, 37°20′12″ N, 112°38′50″ E, 1340 m asl, e.g., *Sanguisorba officinalis* L. (Rosaceae), 31 August 2020, M. Ma and B. Zhang coll.; 15♀ and 3♂, same locality, *Setaria viridis* (Poaceae), 21 August 2021, Y. Liu, M. Ma, B. Zhang and F.-X. Ren coll.12.**Remarks.** *Neoseiulus lushanensis* is classified in the *womersleyi* species subgroup within the *barkeri* species group [37]. Originally described from grass in Jiangxi Province, it has also been recorded in Guizhou, Henan, Hunan, Shandong, and Zhejiang provinces [6]. This paper reports the first record of it in Shanxi Province, where we observed a variation in the length of seta *Z1*. Compared to specimens from Jiangxi (23.4) [15] and Guizhou (26.5–27.5) [6], those from Shanxi have a longer *Z1* (35 (28–36)). In earlier descriptions, the striation on the anterolateral margins of the dorsal shield were often overlooked, as was the case in the original description provided by Zhu and Chen (1985) and the subsequent redescription presented by Wu et al. (2009). It is worth noting that the original description may have presented an incorrect number of preanal setae in males, which should be six pairs instead of three pairs.

#### 3.1.3. *Neoseiulus makuwa* (Ehara, 1972)

*Amblyseius* (*Amblyseius*) *makuwa* Ehara, 1972: 154.

*Amblyseius makuwa* (Ehara), Wu et al., 1991 [39]: 147; Wu et al., 1997: 99 [40].

*Amblyseius* (*Neoseiulus*) *makuwa* (Ehara), Ehara and Amano, 1998: 37 [41]; Wu et al. 2009: 151 [6].

*Neoseiulus makuwa* (Ehara), Moraes et al., 1986: 87 [42]; Chant and McMurtry, 2003: 33 [37]; Moraes et al., 2004: 131 [38].

**Material examined.** A total of 1♀, Dabaishi Village, Taigu County, Shanxi Province, 37°20′12″ N, 112°38′50″ E, 1340 m, e.g., *Setaria viridis* (Poaceae), 31 August 2020, Y. Liu, M. Ma, B. Zhang and F.-X. Ren coll.**Remarks.** The species was first described from *Cucumis melo* L. var. *makuwa* Makino in Kyushu, Japan, by Ehara (1972), and has since been reported in six other Asian countries including China, as well as one African country [3]. The current study documents the first record of this species in Shanxi Province.

#### 3.1.4. *Neoseiulus paraki* (Ehara, 1967) (Figure 12, Figure 13, Figure 14 and Figure 15)

*Amblyseius* (*Amblyseius*) *paraki* Ehara, 1967: 216; Ehara and Yokogawa, 1977: 52 [43]; Ehara et al., 1994: 126 [44].

*Amblyseius* (*Neoseiulus*) *paraki* (Ehara), Ehara and Amano, 1998: 35 [41].

*Neoseiulus paraki* (Ehara), Moraes et al., 1986: 92 [42]; Chant and McMurtry, 2003: 23 [37]; Moraes et al., 2004: 137 [38].

*Typhlodromip paraki* (Ehara), Ehara and Amano, 2004: 9 [45]; Ryu, 2013: 292 [46].

This is the first record for China.

**Diagnosis (female)**. Dorsal shield elongate oval, strongly reticulate, bearing 17 pairs of setae, 16 pairs of lyrifissures and 7 pairs of solenostomes, all smooth except *Z5* serrated; *s4*, *Z4* and *Z5* longer than others. Peritremes extending anteriorly to bases of *j1*. Sternal shield reticulated, bearing three pairs of setae. Ventrianal shield approximately pentagonal, loosely reticulated, solenostomes (*gv3*) small and rounded, posterior to *JV2*. Calyx of spermathecal apparatus bell-shaped and basally stalked, stalk approximately twice as long as width of atrium; atrium broadened at junction with minor duct, minor duct thread-like; major duct narrower than atrium. Fixed digit of chelicera with four–five teeth and movable digit edentate. Palpgenu with genu setae *al1* and *al2* rod-like. Genu II with eight setae. Genu, tibia and basitarsus of leg IV each with a macroseta.**Redescription**. **Female** (n = 3). Dorsal idiosoma (Figure 12A and Figure 13A). Idiosomal setal pattern 10A:9B/JV-3:ZV. Dorsal shield elongate oval, strongly reticulate, with a waist at level of *R1*, 404 (375–404) long, 198 (178–198) wide; muscle marks visible between *j3* and *Z4*. Dorsum with 17 pairs of setae, 16 pairs of lyrifissures (*id1*, *id2*, *id4*, *id6*, *idx*, *idx1*, *idl2*, *idl3*, *idl4*, *idm1*, *idm2*, *idm3*, *idm4*, *idm5*, *idm6* and *is1*) and 7 pairs of solenostomes (*gd1*, *gd2*, *gd4*, *gd5*, *gd6*, *gd8* and *gd9*)); lyrifissures *id3* and solenostomes *gd3* on peritremal shield. All dorsal setae smooth except *Z5* serrated; *Z4* and *Z5* longer than others. Lateral setae *r3* and *R1* smooth, on interscutal membrane. Peritremes extending anteriorly to bases of *j1*, posterior part of peritremal shield (Figure 12B) curved and bluntly pointed, protuberance of exopodal shield situated at level of stigmata. Lengths of setae: *j1* 26, *j3* 33 (32–35), *j4* 19 (18–19), *j5* 16 (16–18), *j6* 22 (21–25), *J2* 25 (24–26), *J5* 12 (12–14), *r3* 29 (29–32), *R1* 27 (27–35), *s4* 41 (41–44), *S2* 39 (38–42), *S4* 34 (33–38), *S5* 34 (33–37), *z2* 28 (28–29), *z4* 30 (30–33), *z5* 19 (19–21), *Z1* 26 (26–29), *Z4* 42 (42–49), *Z5* 65 (65–70).

**Figure 12 animals-13-01478-f012:**
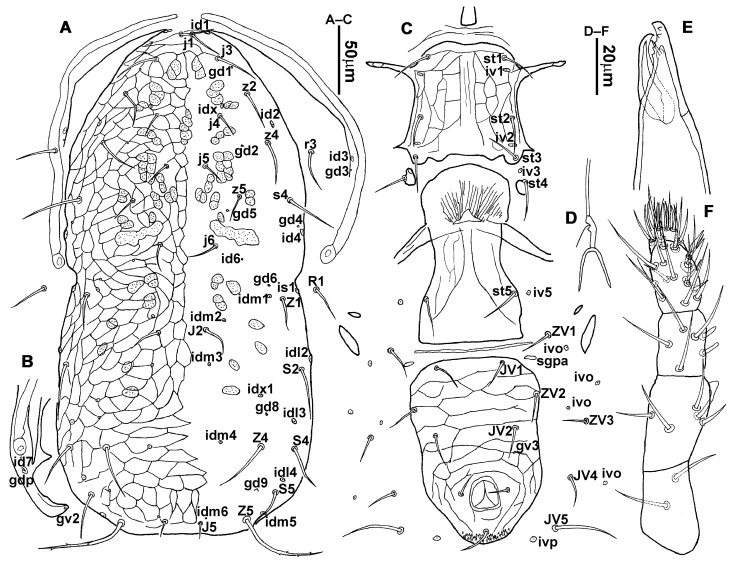
*Neoseiulus paraki* (Ehara), female. (**A**) Dorsal shield; (**B**) Posterior part of peritremal shield and exopodal shield; (**C**) Ventral idiosoma; (**D**) Spermathecal apparatus; (**E**) Chelicera; (**F**) Palp.

3.Ventral idiosoma (Figure 12C and Figure 13B). Sternal shield reticulated, 81 (76–81) long, 85 (80–85) wide; anterior margin convex, forming a flat M-shaped projection, posterior margin weakly concave, with two small lateral projections; three pairs of setae (*st1*, *st2* and *st3*) and two pairs of lyrifissures (*iv1* and *iv2*) present on sternal shield, *iv1* positioned posteriad of *st1*, *iv2* positioned between *st2* and *st3*, and closer to *st3* than to *st2*. Metasternal platelets small, each bearing a seta *st4* and a lyrifissure *iv3*. Epigynal shield slightly striated, 136 (126–136) long, 85 (71–85) wide. Lengths of setae: *st1* 29 (29–30), *st2* 25 (25–27), *st3* 28 (26–28), *st4* 30 (27–30), *st5* 25 (25–27). A slender transverse sclerite present between epigynal and ventrianal shields. Ventrianal shield (Figure 14A,B) approximately pentagonal, loosely reticulated throughout, 145 (135–145) long, 111 (97–111) wide, bearing three pairs of preanal setae (*JV1*, *JV2* and *ZV2*), a pair of paranal setae (*PA*) and a postanal seta (*PST*), and a pair of solenostomes (*gv3*) posterior to *JV2*, *gv3* small and round, distance *gv3*–*gv3* 53 (48–53); four pairs of setae (*JV4*, *JV5*, *ZV1* and *ZV3*) and five pairs of lyrifissures present on soft cuticle surrounding ventrianal shield. A pair of tiny platelets (*sgpa*) posteroparaxial to *ZV1* close to anterior corners of ventrianal shield. Lengths of setae: *JV1* 21 (18–21), *JV2* 25 (22–25), *JV4* 20 (19–22), *JV5* 45 (40–48), *ZV1* 22 (20–22), *ZV2* 22 (21–22), *ZV3* 19 (17–19). Primary metapodal platelet 25 (23–25) long, 5 (5–6) wide; secondary platelet 13 (11–13) long, 2 (2–3) wide.

**Figure 13 animals-13-01478-f013:**
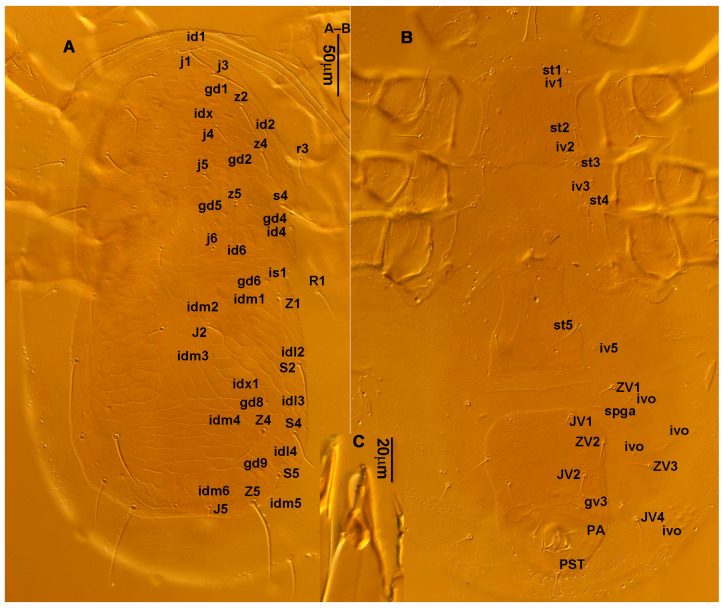
*Neoseiulus paraki* (Ehara), female. (**A**) Dorsal shield; (**B**) Ventral idiosoma; (**C**) Chelicera.

4.Spermatheca (Figure 12D and Figure 14C). Calyx of spermathecal apparatus 19 (18–20) long, bell-shaped and basally stalked, stalk approximately twice as long as width of atrium, atrium broadened at junction with minor duct, minor duct thread-like; major duct approximately half width of atrium.5.Gnathosoma. Chelicera (Figure 12E and Figure 13C) with fixed digit 34 long, bearing four–five teeth, pilus dentilis located at the level of fourth tooth, 5 (5–6) long; movable digit 33 (31–33) long, without teeth. Palp (Figure 12F). Trochanter with two setae; femur with a spatulate and four simple setae; genu bearing two rod-like setae (*al1* and *al2*) and four simple setae; tarsal apotele two-tined.

**Figure 14 animals-13-01478-f014:**
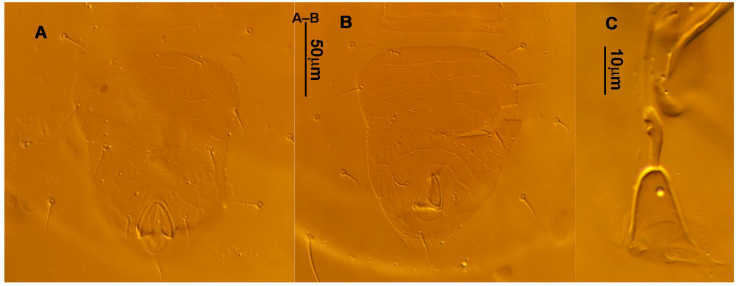
*Neoseiulus paraki* (Ehara), female. (**A**,**B**) Variation in ventrianal shield; (**C**) Spermathecal apparatus.

**Figure 15 animals-13-01478-f015:**
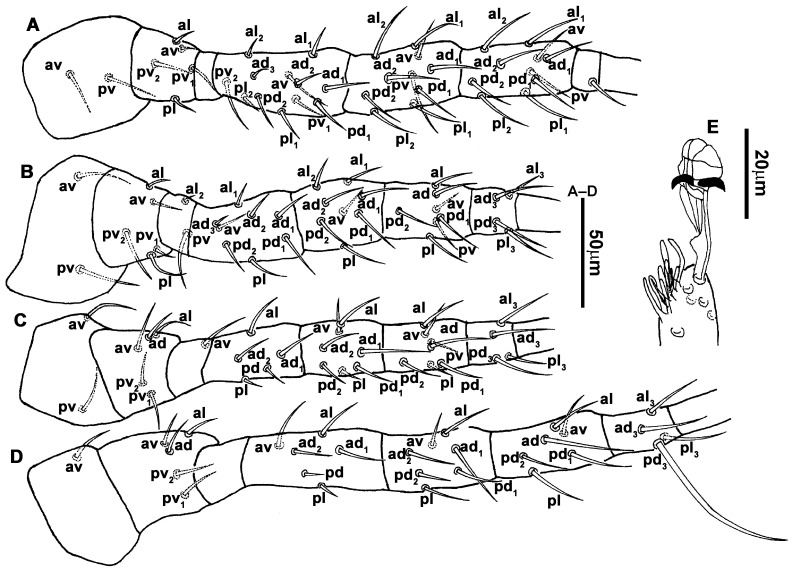
*Neoseiulus paraki* (Ehara), female. (**A**) Leg I; (**B**) Leg II; (**C**) Leg III; (**D**) Leg IV; (**E**) Apical sensorial setal cluster of tarsus I.

6.Legs (Figure 15A–D). Leg I 392 (374–406) long, setal formula: coxa 0-0/1-0/1-0, trochanter 1-0/1-0/2-1, femur 2-3/1-2/2-2, genu 2-2/1-2/1-2, tibia 2-2/1-2/1-2, basitarsus 0-0/0-1/0-0. Apical sensorial setal cluster of tarsus I (Figure 15E) with 10 modified setae. Leg II 323 (311–333) long, setal formula: coxa 0-0/1-0/1-0, trochanter 1-0/1-0/2-1, femur 2-3/1-2/1-1, genu 2-2/1-2/0-1, tibia 1-1/1-2/1-1, basitarsus 1-1/0-1/0-1. Leg III 324 (314–333) long, setal formula: coxa 0-0/1-0/1-0, trochanter 1-1/1-0/1-1, femur 1-2/1-1/0-1, genu 1-2/1-2/0-1, tibia 1-1/1-2/1-1, basitarsus 1-1/0-1/0-1. Leg IV 427 (419–451) long, setal formula: coxa 0-0/1-0/0-0, trochanter 1-1/1-1/1-0, femur 1-2/1-1/0-1, genu 1-2/1-2/0-1, tibia 1-1/1-2/0-1, basitarsus 1-1/0-1/0-1. Legs I–II without macrosetae, genu and tibia of leg III each with a smooth macroseta, *SgeIII* 31 (29–32), *StiIII* 27 (25–28); genu, tibia and basitarsus of leg IV each with a smooth macroseta, *SgeIV* 40 (39–40), *StiIV* 39 (39–41), *StIV* 78 (70–78).7.**Males**. Not found in the current study.8.**Materials examined.** A total of 3♀, Dabaishi Village, Taigu County, Shanxi Province, 37°20′12″ N, 112°38′50″ E, 1340 m, e.g., *Setaria viridis* (L.) P. Beauv. (Poaceae), 31 August 2020, Y. Liu, M. Ma, B. Zhang and F.-X. Ren coll.9.**Remarks.** The female of this species was originally described from apple trees in Sapporo, Hokkaido, Japan [16]. Subsequently, Ehara and Yokogwa (1977) provided information on the male morphology. Both sexes of *N. paraki* have four–five teeth on the fixed digit, while the movable digit is edentate in females but has one tooth in males [43]. While this species has also been reported in South Korea, specimens from this country have seven teeth on the fixed digit and one tooth on the movable digit, and their epigynal shield is strongly reticulated [46]. In contrast, the epigynal shield of our specimens is slightly striated. One of our specimens exhibited a structural anomaly where *iv3* was positioned off the metasternal platelet on the right side.

#### 3.1.5. *Neoseiulus tauricus* (Livshitz and Kuznetsov, 1972) (Figure 16, Figure 17, Figure 18 and Figure 19)

*Amblyseius tauricus* Livshitz and Kuznetsov, 1972: 24; Wu and Lan, 1991: 316 [47]; Wu and Ou, 1999: 105 [48].

*Amblyseius* (*Neoseiulus*) *tauricus* (Livshitz and Kuznetsov), Wu et al., 2009: 92 [6].

*Neoseiulus tauricus* (Livshitz and Kuznetsov), Moraes et al., 1986: 92 [42]; Chant and McMurtry, 2003: 23 [37]; Moraes et al., 2004: 147 [38].

**Diagnosis (female)**. Dorsal shield elongate oval, reticulated throughout, bearing 17 pairs of setae, 16 pairs of lyrifissures and 7 pairs of solenostomes, all smooth except *Z4* and *Z5*, which were serrated; *Z4* and *Z5* longer than others. Peritremes extending anteriorly to bases of *j1*. Sternal shield striated laterally, bearing three pairs of setae. Ventrianal shield approximately pentagonal, mostly transversally striated and a few oblique striae present between transverse striae; solenostomes (*gv3*) not discernible. Calyx of spermathecal apparatus funnel-shaped and constricted medially, arms apically flaring; atrium positioned right at base of calyx; major duct membranous, gradually expanded after a short distance. Fixed digit of chelicera with four teeth and movable digit with a tooth. Palpgenu with genu setae *al1* and *al2* tapered, spiniform. Genu II with seven setae. Genu, tibia and basitarsus of leg IV each with a macroseta.**Redescription**. **Female** (n = 4). Dorsal idiosoma (Figure 16A and Figure 17A). Idiosomal setal pattern 10A:9B/JV-3:ZV. Dorsal shield elongate oval, reticulated, with a waist at level of *R1*, 347 (328–347) long, 171 (160–171) wide; muscle marks visible between *j1* and *Z4*, 2 pairs of muscle marks present in front of *J5*. Dorsum with 17 pairs of setae and 16 pairs of lyrifissures (*id1*, *id2*, *id4*, *id6*, *idx*, *idx1*, *idl2*, *idl3*, *idl4*, *idm1*, *idm2*, *idm3*, *idm4*, *idm5*, *idm6* and *is1*) and 7 pairs of solenostomes (*gd1*, *gd2*, *gd4*, *gd5*, *gd6*, *gd8* and *gd9*); lyrifissures *id3* and solenostomes *gd3* on peritremal shield. All dorsal setae smooth except *Z4* and *Z5*, which were serrated; *Z4* and *Z5* longer than others. Lateral setae *r3* and *R1* smooth, on interscutal membrane. Peritremes extending anteriorly to bases of *j1*, posterior part of peritremal shield (Figure 16B) nearly straight and with a blunt tip, protuberance of exopodal shield at level of stigmata. Lengths of setae: *j1* 19 (15–19), *j3* 18 (17–19), *j4* 11, *j5* 12 (11–12), *j6* 13 (12–13), *J2* 14 (12–14), *J5* 9 (9–10), *r3* 15 (15–18), *R1* 16 (14–18), *s4* 26 (24–26), *S2* 24 (24–26), *S4* 23 (21–23), *S5* 21 (20–22), *z2* 18 (14–18), *z4* 16 (15–17), *z5* 12 (11–13), *Z1* 19 (15–19), *Z4* 37 (36–37), *Z5* 48 (48–53).Ventral idiosoma (Figure 16C and Figure 17B). Sternal shield striated laterally, 68 (62–68) long, 73 (69–73) wide; anterior margin convex, posterior margin nearly plane; three pairs of setae (*st1*, *st2* and *st3*) and two pairs of lyrifissures (*iv1* and *iv2*) present on sternal shield, lyrifissure *iv1* positioned posteriad of *st1*, *iv2* positioned between *st2* and *st3*, and close to *st3*. Metasternal platelets small, each bearing a seta *st4* and a lyrifissure *iv3*. Epigynal shield with a few longitudinal and oblique striae and two pairs of muscle scars between *st5* and *st5*; 118 (110–120) long, 65 (63–68) wide. Lengths of setae: *st1* 29 (25–29), *st2* 24 (24–26), *st3* 28 (22–28), *st4* 23 (20–23), *st5* 22 (21–23). A slender transverse sclerite present between epigynal and ventrianal shields. Ventrianal shield approximately pentagonal, striated, 122 (110–122) long, 92 (89–92) wide, bearing three pairs of preanal setae (*JV1*, *JV2* and *ZV2*), a pair of paranal setae (*PA*) and a postanal seta (*PST*), solenostomes (*gv3*) not discernible; four pairs of setae (*JV4*, *JV5*, *ZV1* and *ZV3*) and five pairs of lyrifissures present on soft cuticle surrounding ventrianal shield. A pair of tiny platelets (*sgpa*) posteroparaxial to *ZV1* adjacent to anterior corners of ventrianal shield. Lengths of setae: *JV1* 20 (19–20), *JV2* 22 (17–22), *JV4* 21 (21–22), *JV5* 50 (50–56), *ZV1* 21 (17–21), *ZV2* 19 (18–20), *ZV3* 11 (11–13). Primary metapodal platelet 27 (26–32) long, 6 (5–6) wide; secondary platelet 13 (11–16) long, 3 (3–4) wide.

**Figure 16 animals-13-01478-f016:**
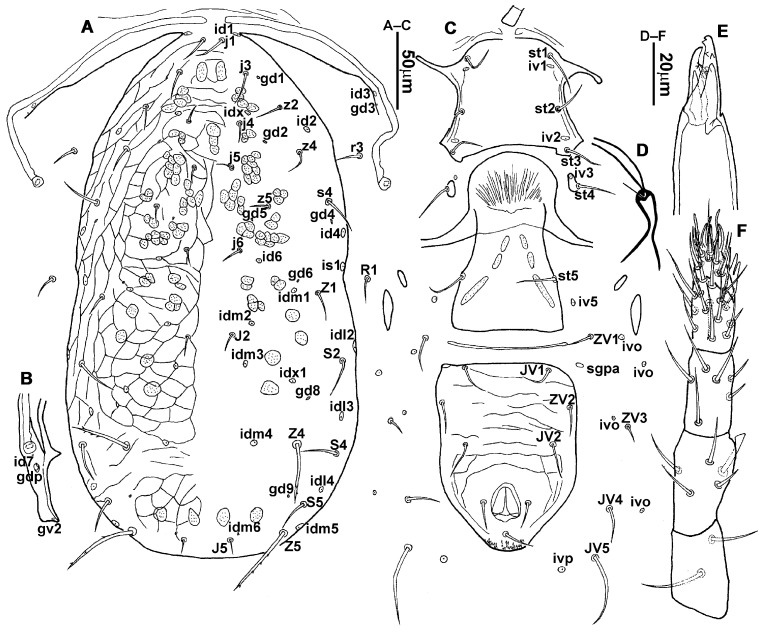
*Neoseiulus tauricus* (Livshitz and Kuznetsov), female. (**A**) Dorsal shield; (**B**) Posterior part of peritremal shield and exopodal shield; (**C**) Ventral idiosoma; (**D**) Spermathecal apparatus; (**E**) Chelicera; (**F**) Palp.

4.Spermatheca (Figure 16D and Figure 18A,B). Calyx of spermathecal apparatus elongate, funnel-shaped and constricted medially, flaring distally, 24 (23–24) long; atrium incorporated within base of calyx; minor duct thread-like; major duct membranous, gradually broadened after a short distance.

**Figure 17 animals-13-01478-f017:**
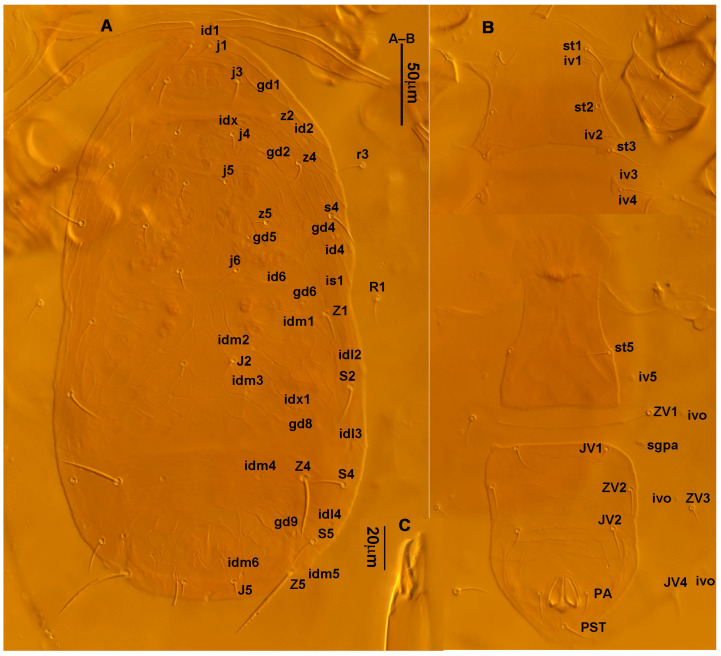
*Neoseiulus tauricus* (Livshitz and Kuznetsov), female. (**A**) Dorsal shield; (**B**) Ventral idiosoma; (**C**) Chelicera.

5.Gnathosoma. Chelicera (Figure 16E and Figure 17C) with fixed digit 27 (27–30) long, bearing four teeth, pilus dentilis located at the level of fourth tooth, 7 long; movable digit 26 (25–26) long, bearing a single tooth. Palp (Figure 16F). Trochanter with two setae; femur with a spatulate and four simple setae; genu bearing two tapered spiniform setae (*al1* and *al2*), and four simple setae; tarsal apotele two-tined.6.Legs (Figure 19A–D). Leg I 339 (327–342) long, setal formula: coxa 0-0/1-0/1-0, trochanter 1-0/1-0/2-1, femur 2-3/1-2/2-2, genu 2-2/1-2/1-2, tibia 2-2/1-2/1-2, basitarsus 0-0/0-1/0-0. Apical sensorial setal cluster of tarsus I (Figure 19E) with 10 modified setae. Leg II 268 (253–268) long, setal formula: coxa 0-0/1-0/1-0, trochanter 1-0/1-0/2-1, femur 2-3/1-2/1-1, genu 2-2/0-2/0-1, tibia 1-1/1-2/1-1, basitarsus 1-1/0-1/0-1. Leg III 268 (245–268) long, setal formula: coxa 0-0/1-0/1-0, trochanter 1-1/1-0/1-1, femur 1-2/1-1/0-1, genu 1-2/1-2/0-1, tibia 1-1/1-2/1-1, basitarsus 1-1/0-1/0-1. Leg IV 352 (328–352) long, setal formula: coxa 0-0/1-0/0-0, trochanter 1-1/1-1/1-0, femur 1-2/1-1/0-1, genu 1-2/1-2/0-1, tibia 1-1/1-2/0-1, basitarsus 1-1/0-1/0-1. Legs I–III without macrosetae. Genu, tibia and basitarsus of leg IV each with a smooth macroseta, *SgeIV* 30 (30–33), *StiIV* 21 (20–23), *StIV* 49 (48–50).7.**Males**. Not found in the present study.

**Figure 18 animals-13-01478-f018:**
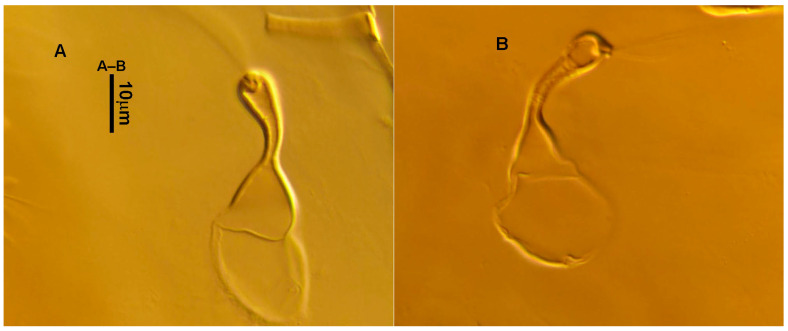
*Neoseiulus tauricus* (Livshitz and Kuznetsov), female. (**A**,**B**) Variation in spermathecal apparatus.

**Figure 19 animals-13-01478-f019:**
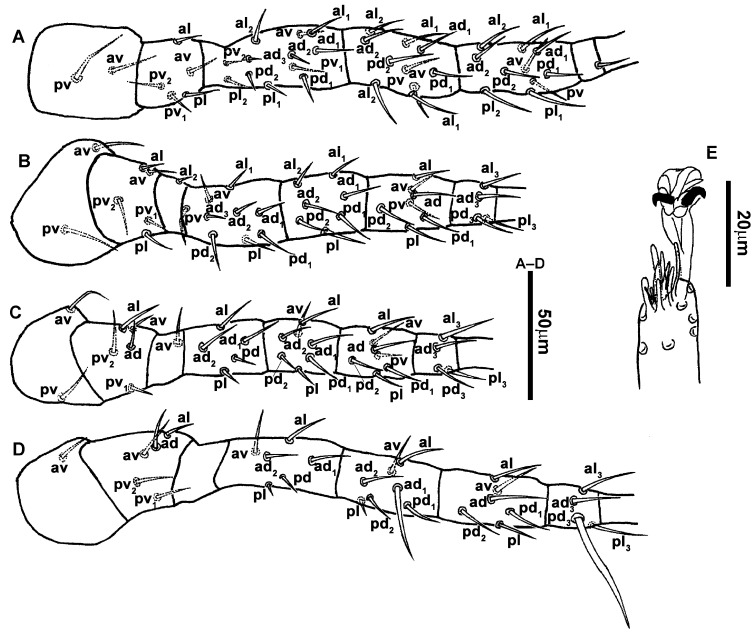
*Neoseiulus tauricus* (Livshitz and Kuznetsov), female. (**A**) Leg I; (**B**) Leg II; (**C**) Leg III; (**D**) Leg IV; (**E**) Apical sensorial setal cluster of tarsus I.

8.**Materials examined.** A total of 4♀, Dabaishi Village, Taigu County, Shanxi Province, 37°20′12″ N, 112°38′50″ E, 1340 m, e.g., *Setaria viridis* (L.) P. Beauv. (Poaceae), 31 August 2020, Y. Liu, M. Ma, B. Zhang and F.-X. Ren coll.9.**Remarks.** *Neoseiulus tauricus* was first described by Livshitz and Kuznetsov (1972). In China, it was previously recorded from weeds in Inner Mongolia [47]. We observed that seta *JV5* in our specimens is smooth, whereas it was originally described as serrated [18]. Additionally, we noted that the left lyrifissure *iv3* is off the metasternal platelet. The macroseta on tibia IV in our Shanxi specimens is indistinct, as reported in [6,40], whereas it is only slightly shorter than that on genu IV in Livshitz and Kuznetsov (1972).

#### 3.1.6. *Neoseiulus womersleyi* (Schicha, 1975)

*Amblyseius womersleyi* Schicha, 1975: 101; Schicha, 1987: 96 [49].

*Amblyseius* (*Amblyseius*) *womersleyi* (Schicha), Tseng, 1983: 54 [50]; Ehara et al., 1994: 123 [44].

*Amblyseius* (*Neoseiulus*) *womersleyi* (Schicha), Ehara and Amano, 1998: 30 [41]; Wu et al., 2009: 65 [6].

*Neoseiulus womersleyi* (Schicha), Moraes et al., 1986: 86 [42]; Beard, 2001: 84 [51]; Chant and McMurtry, 2003: 37 [37]; Moraes et al., 2004: 152 [38]; Liao et al., 2020: 288 [52].

*Amblyseius pseudolongispinosus* Xin, Liang and Ke, 1981: 75 [9]; Wu et al. 1997: 43 [40]; synonymy by Tseng, 1983: 57 [50].

*Amblyseius* (*Neoseiulus*) *pseudolongispinosus* (Xin, Liang and Ke), Wu et al., 2009: 65 [6].

*Neoseiulus pseudolongispinosus* (Xin, Liang and Ke), Ma et al., 2015: 15 [10].

**Materials examined.** A total of 2♀, Jincheng City, Shanxi Province, 35°29′31″ N, 112°49′47″ E, 624 m, e.g., weed, 19 August 2014, Y.-N. Zhao coll.; 1♂, Shanxi Agriculture University, Taigu County, Shanxi Province, 37°25′24″ N, 112°34′53″ E, 794 m, weed, 20 September 2014, M.-J. Yi and B.-Q. Su coll.**Remarks.** This species was originally described from strawberry in Australia [8] and later recorded in China, Japan and South Korea [3]. In 1981, Xin et al. described a new species, *Amblyseius pseudolongispinosus*, based on specimens collected from various plant species in different regions of China. This species has been reported in Mainland China under the name *Amblyseius pseudolongispinosus*, although Tseng, from Taiwan, synonymized it with *Amblyseius womersleyi* in 1983. Our examination of specimens from different localities in China, as well as the original description and specimens from New Zealand, supports Tseng’s decision to synonymize the two species.

#### 3.1.7. *Neoseiulus zwoelferi* (Dosse, 1957)

*Typhlodromus zwoelferi* Dosse, 1957: 301 [53].

*Cydnodromus zwoelferi* (Dosse), Muma, 1961: 290 [54].

*Amblyseius zwoelferi* (Dosse), Schuster and Pritchard, 1963: 268 [55].

*Amblyseius* (*Amblyseius*) *zwoelferi* (Dosse), Wainstein, 1975: 920 [56].

*Amblyseius subreticulatus* Wu, 1987: 264 [57]; synonym by Zhang et al. 2021: 20 [12].

Third bullet.

**Materials examined.** A total of 17♀, 5♂, 8 deutonymph, 8 protonymphs, 6 larvae, e.g., laboratory culture in Shanxi Agriculture University, 8 June 2020–23 September 2020, B. Zhang, M. Ma and S. Jiao. coll.; 5♀, Taigu, e.g., *Hemerocallis fulva* L. (Asphodelaceae), 7 October 2013, Q.-H. Fan coll.; 1♀, Taigu, e.g., *Hemerocallis fulva* L. (Asphodelaceae), 6 August 2014, M. Ma coll.; 17♀, 2♂, 1 protonymph. Ningwu, Luyashan National Nature Reserve, e.g., weed, 6 August 2014, B. Zhang and M. Ma coll.; 1♀, Ningwu, Luyashan National Nature Reserve, e.g., *Hippophae rhamnoides* L. (Elaeagnaceae), B.-Q. Su and M.-J. Yin coll.; 1♀, 1♂, Taigu, e.g., weed, 20 August 2014, B.-Q. Su and M.-J. Yin coll.**Remarks.** The original description of this species was based on specimens collected from apple leaves in Oldenburg, Germany [53]. Since then, it has been found in eighteen countries in the Palearctic realm, one country in the Indomalayan realm, and one country in the Nearctic realm [3]. In China, it has a broad geographical range, being present from the far northeast (Heilongjiang) to the far northwest (Xinjiang) and the southern coast (Guangdong).

### 3.2. Key to Adult Females of Neoseiulus in Shanxi Province

1. Most dorsal idiosomal setae (except j1 and J5) serrated; setae *j4*–*6*, *Z1*, *S2* and *S4* extending beyond bases of setae in next row. ………….… *N. womersleyi* (Schicha)

- Most dorsal idiosomal setae smooth, except *Z5*, which is barbed, *Z4*, S4 and S5, which are barbed or smooth; setae *j4*–*6*, *Z1*, *S2* and *S4* not extending beyond bases of setae in the next row. ……………………………………………….……………… 2

2. Atrium of spermathecal apparatus bifurcated at junction with major duct; calyx elongate trumpet-shaped; spermatodactyl of male T-shaped. …………………… 3

- Atrium of spermathecal apparatus not bifurcated at junction with major duct; calyx does not elongate trumpet-shaped; spermatodactyl of male L-shaped. ….… 4

3. Setae *z4*, *s4* and *Z4* long, extending beyond or nearly reaching to bases of setae in next row; macrosetae present on genu, tibia and basitarsus of leg IV. ……………. …………………………………………………………. *N. lushanensis* (Zhu and Chen)

- Setae *z4*, *s4* and *Z4* short, far from bases of setae in next row; macrosetae only on genu and basitarsus of leg IV. ……………………………………. *N. makuwa* (Ehara)

4. Calyx of spermathecal apparatus not stalked, immediately attached to calyx; major duct expanded, approximately as wide as medial part of calyx; palpgenu anterior lateral setae *al1* and *al2* tapered, spiniform; solenostomes (*gv3*) absent. ………….. …………..……………………………………… *N. tauricus* (Livshitz and Kuznetsov)

- Calyx of spermathecal apparatus basally stalked; major duct slender; palpgenu setae *al1* and *al2* cylindrical, rod-like; solenostomes (*gv3*) present. ………………. 5

5. Genu II with 7 setae; genu IV without obvious macrosetae; *S4* and *S5* serrated. … …………………………………………………………………... *N. bicaudus* (Wainstein)

- Genu II with 8 setae; genu IV with macroseta at least 1.5 times length of other setae; *S4* and *S5* smooth. …………………………………………………………………. 6

6. Calyx of spermathecal apparatus basally pointed, V-shaped; seta *S4* approximately half distance *S4*–*S5*; *s4* approximately half distance *s4*–*z5*. …… *N. zwoelferi* (Dosse)

- Calyx of spermathecal apparatus basally rounded, U-shaped; *S4* nearly as long as distance *S4*–*S5*; *s4* longer than distance *s4*–*z5*. …………………….. *N. paraki* (Ehara)

## 4. Discussion

*Neoseiulus* is the third most abundant and most widely distributed genus across the globe after *Typhlodromus* and *Amblyseius* in Phytoseiidae. The species of this genus are predominantly found in plants on every continent, except Antarctica, and their largest diversity is concentrated in tropical and subtropical regions. The ancestral habitat of *Neoseiulus* is presumed to be amongst plants that grew close to the ground as well as the ground litter [37]. The species documented above are unexceptional inhabitants of plants, much as most other species in the genus *Neoseiulus*. The green foxtail (*Setaria viridis*), a species that is abundant in Shanxi and other areas in northern China, appears to be an ideal habitat for the *Neoseiulus* in Shanxi, hosting five (*N. bicaudus*, *N. lushanensis*, *N. makuwa*, *N. paraki* and *N. tauricus*) out of the seven species recorded; however, for some species, this is based on a small number of specimens.

Ten species groups are recognized in *Neoseiulus* and two of them contain four species subgroups each, while the others remain undivided [3,37,38]. The species found in Shanxi can be classified into two groups: *barkeri* species group (*N. lushanensis*, *N. makuwa* and *N. womersleyi*) and *cucumeris* species group (*N. bicaudus*, *N. paraki* and *N. zwoelferi* in *paraki* species subgroup and *N. tauricus* in *ceratoni* species subgroup).

The classification of *Neoseiulus* primarily relies on variations in the morphology of the spermatheca [2,6,33,37,49,51]. In addition, other characteristics utilized include the idiosomal setal pattern, dorsal idiosomal setal ratio, ornamentation of the dorsal shield, length-to-width ratio of the sternal shield, the position of sternal seta *st3*, the shape of the female ventrianal shield, the shape and position of solenostomes *gv3*, the number of teeth on the movable and fixed cheliceral digits of females, the number of macrosetae on leg IV, the number of setae on genu II, and the shape of the male spermatodactyl. Taxonomists have continuously searched for new morphological characteristics to improve the systematics of Phytoseiidae. Certain characteristics, such as the shape of the posterior section of the peritremal shield and exopodal shields, the location of gland pores and poroids/lyrifissures on the dorsal and ventral idiosomal shields, as well as on the peritremal shield, in addition to the shape of the specialized setae on femur (*al*) and genu (*al1*, *al2*), and leg chaetotaxy, are presented in a number of publications [2,33,37,51,52,58,59,60,61,62,63]. Further research is necessary to comprehensively evaluate these characteristics in a systematic way.

This is the first review of the genus *Neoseiulus* in Shanxi Province, recording seven species which account for 12.3% of the total known species of this genus in China. Among the findings, *N. paraki* (Ehara), recorded in the current work, was previously only known to be present in Japan and South Korea. Additionally, the study confirms *N. neoreticuloides* (Liang and Hu) as a new junior synonym of *N. bicaudus* (Wainstein). It should be noted that, as there have been no previous surveys of phytoseiid mites in the province, the data presented here may be limited, and it is likely that more species have yet to be discovered.

## 5. Conclusions

The study of *Neoseiulus* in Shanxi Province has added valuable information to the study of phytoseiid fauna in the area. The redescriptions of four species contribute to the enrichment and provide a more detailed basis for species identification. The recording of *N. paraki* in China for the first time expands the known distribution of this species, and synonymizing *N. neoreticuloides* with *N. bicaudus* clarifies the taxonomic status of these species. The diagnostic key provided will assist in the identification of the known species of *Neoseiulus* in Shanxi and will be useful for future studies on the genus. In brief, this study highlights the importance of conducting further investigations into the species diversity and distribution across different geographical locations.

## Figures and Tables

**Table 1 animals-13-01478-t001:** Comparison of morphological characteristics of *Neoseiulus neoreticuloides* and *N. bicaudus*.

	*N. neoreticuloides* [13]	*N. bicaudus* [36]	*N. bicaudus* [28]	*N. bicaudus* [31]	*N. bicaudus* [26]
Dorsal shield	reticulate, 415.7 long, 206.7 wide	thin net-like on posterior part, 400 long, 180 wide	reticulate, 402–410 long, 190–200 wide	reticulate, 388–401 long, 167–202 wide	reticulate, 375–425 long, 160–178 wide
Peritreme	extending to level between *j1* and *j3*	extending forward to level of bases of setae *j1*	extending to level between *j1* and *j3*	extending to level between *j1* and *j3*, close to *j3*	extending to level between *j1* and *j3*, close to *j1*
*j1*	25	25	22–25	23–25	22–26
*j3*	30	31	26–31	28–30	27–32
*j4*	15	15	14	14–16	12–15
*j5*	-	15	13–15	14–15	12–15
*j6*	16	17	16–20	16–19	15–18
*J2*	19.5	21	17–20	17–19	17–22
*J5*	14	14	14–16	12–15	12–15
*r3*	34	34	29–33	29–32	30–35
*R1*	30	30	25–31	28–32	30–32
*s4*	32	34	30–34	30–34	30–37
*S2*	33.5	35	32–38	32–36	33–40
*S4*	37.5	43	44–46	36–40	35–42
*S5*	39.5	45	53–61	42–46	40–49
*z2*	24	23	18–23	21–29	20–26
*z4*	20	20	18–19	13–18	16–20
*z5*	15.5	14	14–15	13–15	12–15
*Z1*	23	20	21–23	21–24	20–25
*Z4*	39	40	35–41	34–38	36–42
*Z5*	83	98	92–99	87–95	82–97
Sternal shield	scarcely striated	smooth	reticulated	scarcely striated	reticulated
Ventrianal shield	approximately triangle	135 long, 117 wide, sub-triangular	135–140 long, 110 wide, approximately triangle	133–141 long, 99–120 wide, approximately triangle	135–145 long, 102–114 wide, approximately triangle
Distance *gv3*–*gv3*	-	-	40–46	-	36–47
*JV5*	serrate; 60	serrate; 63	serrate; 64–78	serrate	serrate; 60 (55–65)
Fixed digit	6 teeth	6 teeth	6 teeth	6 teeth	7 teeth
Movable digit	1 tooth	1 tooth	1 tooth	1 tooth	2 teeth
Calyx of spermathecal apparatus	bowl-shaped	cup-shaped	bowl-shaped	bowl-shaped	bell-shaped
Macroseta *StIV*	74	73 (73–78)	64–74	70–73	67–85

Note: “***-***” indicates that the article does not describe this feature.

## Data Availability

The data presented in this study are available on request from the corresponding author.

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
