# Peer review of "The Genus Neoseiulus Hughes (Acari: Phytoseiidae) in Shanxi, China†"

_animals, 2023, doi:10.3390/ani13091478_

Round 1

Reviewer 1 Report

Dear Authors

The manuscript "The genus Neoseiulus Hughes (Acari: Phytoseiidae) in Shanxi, China" is an important contribution to the knowledge of the Phytoseiidae fauna in China, mainly in Shanxi, a province with few records of mites of this family. This manuscript presents new records for Shanxi, as well as for China, with morphological information on some of these recorded species. The manuscript is very well written. The authors present several morphological information, some rarely treated in other works of this family. The illustrations are very well detailed, with also inclusion of microscopy images. I only believe that the authors could include the average measurement of the specimens studied. Small corrections, in the attached file.

Reviewer 2 Report

This manuscript can be published in your journal after the following minor comments are accounted for.

-       Leg chaetotaxy. The chaetotaxy formulae of trochanter III and IV are not correct because of the position of some setae as illustrated by the authors. For detail, please see our comments in the manuscript.

-       When comparing N. neoreticuloides and N. bicaudus, please use Kolodochka 2018 “Two new species of the genus Neoseiulus (Parasitiformes, Phytoseiidae) with redescriptions of N. bicaudus and N. micmac based on holotypes” (doi 10.2478/vzoo-2018-0031).

-       For other minor comments see the pdf file.

Please note that I used  the Preview program for Mac OS to make comments in the pdf file

Reviewer 3 Report

This paper only needs minor revision. Excellent job.

Reviewer 4 Report

This paper is very good and will be certainly very useful for all researchers on the family Phytoseiidae, especially for those from China and also for biodiversity specialists. Redescriptions are very complete, very informative compared to previous descriptions and drawings are excellent.

However, I have detected a lot of small errors that must be corrected, especilly a lack of several more or less important information.

My main concerns are that you claim for synonymy but nothing is said about type material comparisons between the two species involved. If you have done those comparisons, you must tell to readers in material and methods where types re coming from, which one were used, etc., give clearly results and acknowledge for type loans.

If no, you must tell to the readers why you have not done these type comparisons, what are exactly your arguments. I agree with authors on the probable synonymy ! But many authors will not accept this synonymy without type comparisons and they will claim that nothing is demonstrated. So you must give arguments if you have not done and be more carefull in assertions, talk about very probable synonymy for example.

You must also add some arguments, justifications of all your choices. Many are even not developed at all in the text. 

All details are in the manuscript here attached.
